# Identification of the best housekeeping gene for RT-qPCR analysis of human pancreatic organoids

Alessandro Cherubini , Francesco Rusconi, Lorenza Lazzari*

Department of Transfusion Medicine and Haematology, Laboratory of Regenerative Medicine—Cell Factory, Fondazione IRCCS Ca' Granda Ospedale Maggiore Policlinico, Milan, Italy

⊘ These authors contributed equally to this work.
* lorenza.lazzari@policlinico.mi.it

## Abstract

In the last few years, there has been a considerable increase in the use of organoids, which is a new three-dimensional culture technology applied in scientific research. The main reasons for their extensive use are their plasticity and multiple applications, including in regenerative medicine and the screening of new drugs. The aim of this study was to better understand these structures by focusing on the choice of the best housekeeping gene (HKG) to perform accurate molecular analysis on such a heterogeneous system. This feature should not be underestimated because the inappropriate use of a HKG can lead to misleading data and incorrect results, especially when the subject of the study is innovative and not totally explored like organoids. We focused our attention on the newly described human pancreatic organoids (hPOs) and compared 12 well-known HKGs (ACTB, B2M, EF1α, GAPDH, GUSB, HPRT, PPIA, RNA18S, RPL13A TBP, UBC and YWHAZ). Four different statistical algorithms (NormFinder, geNorm, BestKeeper and ΔCt) were applied to estimate the expression stability of each HKG, and RefFinder was used to identify the most suitable genes for RT-qPCR data normalization. Our results showed that the intragroup and intergroup comparisons could influence the best choice of the HKG, making clear that the identification of a stable reference gene for accurate and reproducible RT-qPCR data normalization remains a critical issue. In summary, this is the first report on HKGs in human organoids, and this work provides a strong basis to pave the way for further gene analysis in hPOs.

## Introduction

In the last few years, there has been a remarkable increase in the use of organoid technology in scientific research. Organoids are three-dimensional cellular bodies that can be generated from pluripotent stem cells (embryonic or induced) or adult tissue-specific ancestor cells [1]. The main reasons for their considerable use are their plasticity and multiple applications, including as new models to study specific diseases [2], in regenerative medicine [3] and the screening of new drugs [4].

**Data Availability Statement:** All relevant data are within the paper and its Supporting Information files.

**Funding:** This work was funded by the grant "LSFM4LIFE-Production and characterization of

endocrine cells derived from human pancreas organoids for the cell-based therapy of Type 1 diabetes" project number 668350. The funders had no role in study design, data collection and analysis, decision to publish, or preparation of the manuscript.

**Competing interests:** The authors declare no competing interests.

Many types of organs can be recapitulated in various types of organoids, including the gut [5], brain [6], kidney [7] and liver [8]. Human organoids have been widely studied for a variety of purposes. In particular, human pancreatic organoids (hPOs) have been characterized and presented to be a potential source of functional cells for the treatment of type 1 diabetes [9, 10].

For any type of hPO application such as a three-dimensional platform for tissue regeneration, disease modelling, and drug screening, the gene expression of specific tissue-related genes is a crucial requisite to define their identity. To monitor gene expression, RT-qPCR is often the method of choice due to its sensitive and specific detection, potential for high throughput, rapid and accurate quantification, and high degree of potential automation. In order to obtain gene expression results that are not only accurate but also comparable among different experimental setups, conditions, operators and laboratories, normalization of RT-qPCR data should be performed against one or two housekeeping genes (HKGs). These HKGs must display unchanged cellular expression, irrespective of the experimental conditions, and this is fundamental to achieve reliable results [11].

Moreover, the selection of the HKG for normalization of RT-qPCR data should take into account the expression stability of HKGs among different specimens and experimental conditions [11, 12]. This point should not be underestimated because an inappropriate choice of the HKG can lead to misleading data and incorrect results [13, 14], especially when the subject of the study is innovative and not totally explored like organoids. Unfortunately, the importance of selecting appropriate reference genes is not adequately emphasized by researchers; therefore, the validation process and the comparison of molecular biology data remain controversial and open topics.

The aim of this study was to compare 12 well-known HKGs (ACTB, B2M, EF1α, GAPDH, GUSB, HPRT, PPIA, RNA18S, RPL13A TBP, UBC, and YWHAZ; S1 Table), belonging to different functional families, on hPOs at different passages (P0, P2, P5, P7, P10) in order to identify the most stable HKGs suitable for hPO analysis. Finally, we quantified the differential expression of a panel of pancreatic markers to assess the effect of HKG variability on their transcriptional patterns.

This is the first report on HKGs in human organoids, and this work provides a strong basis to pave the way for further gene analysis in hPOs.

## Materials and methods

### hPO isolation and culture

hPOs were generated starting from adult healthy islet-depleted pancreatic tissue (gently provided by the Diabetes Research Institute, IRCCS Ospedale San Raffaele, Milan, Italy), after approval of the Institutional Review Board (National Transplant Center accredited facility IT000679), as previously described [9]. All experiments were performed according to the amended Declaration of Helsinki. The use of human specimens was approved by the Ethical Committee of Fondazione IRCCS Ca' Granda Ospedale Maggiore Policlinico n˚ 1982, 14th January 2020.

Briefly, pancreatic tissue was dissociated by GentleMACS Dissociator (Miltenyi), then the resulting fragments were embedded into Matrigel (Corning, cat. #356231) and cultured in 24-well non-tissue culture microplates (Sarstedt, cat. #83.3922.500) covered by the complete growing medium: AdDMEM/F12 medium (Gibco, cat. #12634–010) supplemented with 10 mM HEPES (Sigma, cat. #H3375), 1X Glutamax (Gibco, cat. #35050061) 1X Penicillin-Streptomycin (Gibco, cat. 15140122), 1X B27 without vitamin A (Gibco, cat. # 12587–010), 1X N2 supplement (Gibco, cat. # 17502048), 1 mM N-acetyl-L-cysteine (Sigma, cat. #A9165), 1 μg/ml

recombinant human protein Rspo1 (Peprotech, cat. #120–38), 0.1 μg/ml recombinant human Noggin (Peprotech, cat. #120-10C), 50 ng/ml recombinant human EGF (Peprotech, cat. #AF-100-15), 10 nM human (Leu15)-Gastrin I (Sigma, cat. #G9145), 100 ng/ml recombinant FGF10 (Peprotech, cat. #100–26), 10 mM Nicotinamide (Sigma, cat. #N0636), 10 μM Forskolin (Tocris, cat. #1099), 0.5 μM A83-01 (Tocris, cat. #2939) and 3 μM Prostaglandin E2 (Tocris, cat. #2296). The medium was changed every 3 days, and hPOs were split at a ratio of 1:4 every 7 days.

## Flow cytometric analysis

hPOs were collected and washed with fresh AdDMEM/F12 medium supplemented with 10 mM HEPES, 1X Glutamax, 1X Penicillin-Streptomycin and centrifuged for 5 min at 300 x *g* to remove the Matrigel. Then, the cells were dissociated by trypsinization using Tryple (Life Technologies ref. 12604013) for 15 min to obtain a single-cell suspension. We stained the samples using 1:200 FITC mouse Anti-Human Epithelial Cell Adhesion Molecule (EpCAM; BD, cat. #347197), 1:100 Alexa Fluor® 647 Mouse Anti- SRY-box transcription factor 9 (Sox9; BD, cat. #565493) and 1:2000 FITC-conjugated lectin from *Ulex europaeus* (UEA1) (Sigma-Aldrich, cat. #L9006). For surface marker analysis (EpCAM and UEA1), the cells were incubated with fluorophore-conjugated antibodies for 20 min in the dark at room temperature (RT) and then washed with PBS. For intracellular marker analysis (SOX9), the cells were fixed with 4% paraformaldehyde (Electron Microscopy Sciences, ref. 15710) for 30 min at RT and permeabilized for 1 h on ice with permeabilization buffer (PBS + 5% NGS (Sigma-Aldrich, cat #NS02L) and 0.1% TRITON X (Eurobio, cat #GAUTTR00-01). Cells were then stained with primary antibody resuspended in permeabilization buffer and prepared for analysis, as described for the extracellular markers. Finally, the samples were analysed in a FACSCanto II cytometer with FACSDiva analysis software (Becton Dickinson).

## RNA extraction and qPCR analysis

Total RNA extraction was performed using TRIzol reagent (Ambion, cat. #15596–026), according to the manufacturer's instructions. Next, its concentration and quality were measured using a NanoDrop ND-100 spectrophotometer (NanoDrop Technologies), and only samples with 260/280 and 260/230 ratios > 1.8 were accepted. cDNA was synthesized starting from 500 ng of the extracted RNA using iScript Advanced cDNA Synthesis Kit for RT-qPCR (BioRad, cat. #1725038), following the manufacturer's indications. Moreover, to avoid side effects caused by retrotranscription efficiency, we added ERCC RNA Spike-In mix diluted 1:500 to the RNA (Invitrogen, cat. #4456740). The resulting product was diluted 1:10, and 1 μL was used as the template for RT-qPCR. qPCR analysis was completed using SYBR Selected Master Mix (Life Technologies, cat. # 4472908) on a CFX96 thermal cycler (BioRad). All primers used in this study, listed in S2 Table, were designed using Primer 3 software.

## Primer efficiency and quality

The efficiency of each pair of primers was tested. Briefly, serial dilutions starting from 100 ng of cDNA (6 dilutions were generated) of cDNA samples were used as templates for qPCR analysis. The standard (STD) function of the CFX Manager™ Software (Biorad, cat. #1845000) was run to generate the linear regression line, and the efficiency of each pair of primers was obtained. To exclude the generation of nonspecific amplicons, qPCR products were run on an 2% agarose gel with dH$_2$O. The products were isolated from the gel using the Wizard SV Gel and PCR Clean-Up System (Promega, cat. #A9282) and were sequenced to underline the specificity of the primers used by Sanger sequencing (S1 File).

### Best HKG selection

The selection of the best HKG was carried out by focusing on four different methods: ΔCt method (which values the fluctuation of the ΔCt making a comparison between two or more HKGs) [15], geNorm software (www.genorm.cmgg.be; which calculates the stability of each gene through intragroup differences and mean pairwise variation) [16], NormFinder software (www.moma.dk/normfinder-software; which evaluates gene stability using both intragroup and intergroup changes) [17] and BestKeeper software (www.gene-quantification.de/bestkeeper.html; which ranks the genes in agreement with the standard deviation of their Ct values in correlation with intragroup alterations) [18].

The data were integrated to obtain a final rank, based on the geometric mean, using the RefFinder tool (www.github.com/fulxie/RefFinder).

### Statistical analysis

All statistical analyses were performed using Prism7 (GraphPad) on three biological replicates for each experiment. All data passed the Grubb's test excluding the presence of outliers and then were statistically assessed by the non-parametric one-way analysis of variance (ANOVA) test followed by Bonferroni or Dunnett's multiple comparisons post-hoc test, as indicated in the figure legends.

## Results

### hPO characterization

The hPOs showed a cauliflower- or cyst-like structure when they were embedded in Matrigel, and they could be passaged and maintained in culture for at least 10 passages without showing phenotypic changes (Fig 1A). After culturing for 10 days, the cellular composition was characterized. Flow cytometric analysis showed that the majority of the hPO cells (>95%) were positive for EpCAM, thus recapitulating the epithelial exocrine compartment of the initial pancreatic tissue. Focusing on SOX9 and UEA1, well-known markers for ductal and acinar cells, respectively [9, 19, 20], we observed a consistent presence of SOX9-positive cells (71.0% ± 10.0%) and a lower percentage of UEA1-positive cells (19.3% ± 5.0%) (Figs 1B and S1).

### Expression profile of candidate HKGs

To determine the most reliable reference genes, 12 widely used HKGs, belonging to different functional classes or pathways, were selected to reduce the probability of including co-

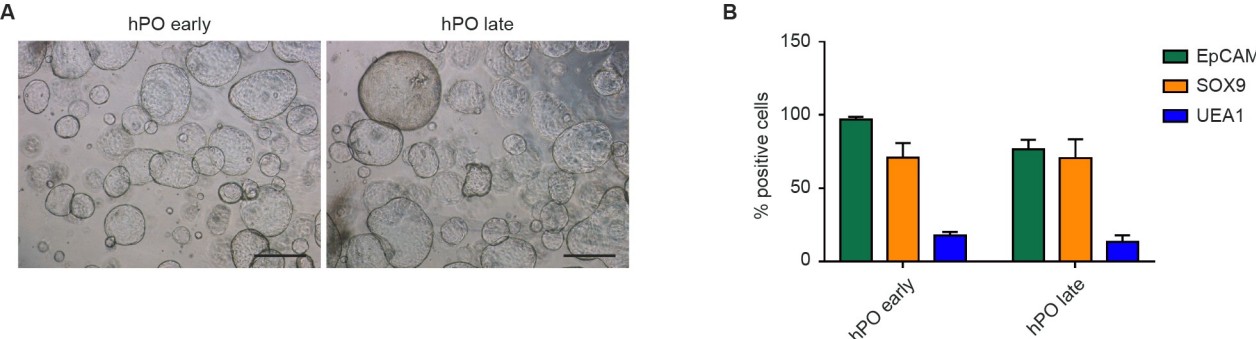

**Fig 1. Human pancreatic organoid characterization.** (A) Representative bright-field images taken after organoid culture of early (< 2 months) and late (> 2 months) passages for 10 days. Scale bar, 500 µm. (B) Immunophenotypic analysis of hPOs at different passages by evaluation of ductal and acinar markers. Data are expressed as the mean ± standard deviation (n = 3/group). One-way analysis of variance followed by the Bonferroni post-hoc test for multiple comparisons was used.

regulated genes (S1 Table). Agarose gel electrophoresis of the RT-qPCR products showed that all primer pairs amplified products of the predicted size, without evidence of unspecific amplicons in either the cDNA or negative control reaction (Fig 2A). Moreover, all products showed individual dissociation curves at a temperature higher than 75˚C, indicating that they were not primer dimers (Fig 2B). In addition, a standard curve was generated for each HKG using two-fold serial dilution of cDNA. The curves showed that the RT-qPCR efficiency of each primer pair ranged between 95% and 105% (S2 Table), which is in agreement with MIQE guidelines [11]. Considering all samples and setting a Ct cut-off > 35 for weak expression, the 12 candidate reference genes exhibited a wide range of expression levels: RNA18S (Ct = 9.73 ± 1) had the highest expression and YWHAZ (Ct = 31.88 ± 1.24) had the lowest expression (Fig 3A). For each reference gene, all Ct values showed that any outliers passed the Grubb's test; thus, no sample was excluded from further analyses. Finally, pairwise comparison between all samples under investigation (all passages analysed) showed a distribution with a correlation coefficient ≥0.96, indicating high expression similarity (Fig 3B).

## Determination of HKG expression stability

With the aim of defining the most stable reference genes, the expression stability of the selected HKGs was determined considering all selected passages (Fig 3C). The Mann–Whitney test was performed for each reference gene. The results indicated that B2M (P0 vs P5), ACTB (P0 vs P5 and P2 vs P5), GAPDH (P0 vs P7), GUSB (P2 vs P7), YWHAZ (P2 vs P10), PPIA (P2 vs P10 and P7 vs P10) had a significantly different expression (P < 0.05) (Table 1). Moreover, a low intragroup variability in our HKGs expression (Coefficient of variation (CV) < 5%) was observed with the exception of ACTB (CV < 10%) and RNA18S (CV < 15%) confirming the stability of selected HKGs (S3 Table).

The stability ranking for each category was evaluated using four different statistical analyses, NormFinde, geNorm, BestKeeper and ΔCt method, as reported in Table 2. Moreover, since we used various approaches showing differences in the HKG stability, a comprehensive final ranking of the most stable reference genes, based on the geometric mean (geo.mean) of each gene weight generated by the four methods, was obtained using the RefFinder tool. Considering all the passages analysed the best housekeeping gene to normalize a RT-qPCR study on human pancreatic organoids were RPL13A (geo.mean = 1.19), and HPRT (geo. mean = 1.73) whereas the least stable HKG were RNA18S (geo.mean = 10.49) and ACTB (geo. mean = 11.24) (Fig 4).

## Impact of HKGs on the expression levels of selected genes

To further evaluate the impact and reliability of the selected reference genes, RT-qPCR analysis was conducted on two specific marker genes for epithelial ductal cells (EpCAM and SOX9), which represent the most enriched cell population in hPO culture. Next, the expression levels were normalized using both the most (RPL13A), the least (ACTB) stable HKGs and exogenous RNA (ERCC RNA Spike-In) as alternative normalization approach, due to their quality and quantity is known [21–23]. Moreover, to avoid possible errors related to the use of only one HKG [11], we also normalized the results using the geo.mean of the Ct value of the two best reference genes identified by NormFinder (PPIA and UBC). Finally, we performed the gene expression normalization using GAPDH, the widely used HKG in the literature. After data normalization, we evaluated the modulation of gene expression using hPO P0 as a reference (Fig 5). We observed a stable expression of EpCAM and SOX9 only when our selected HKG (RPL13A) and the geo.mean were used, on the other hand, using ERCC RNA Spike-In or GAPDH to perform the normalization,

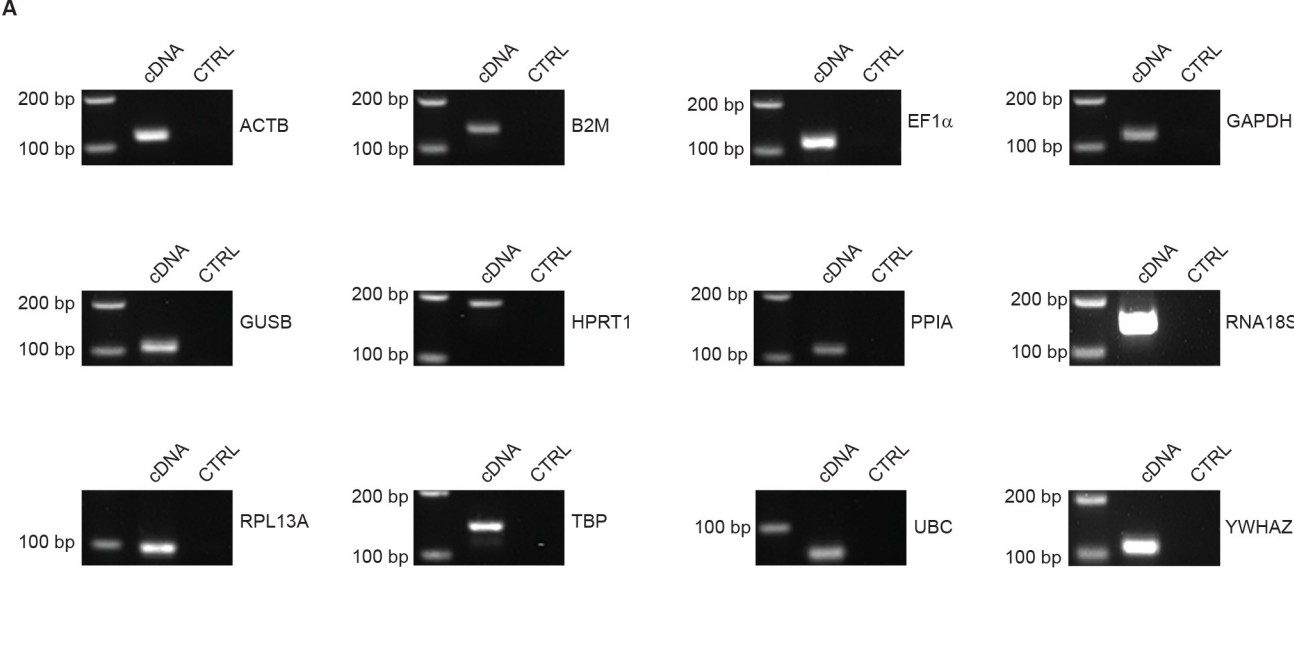

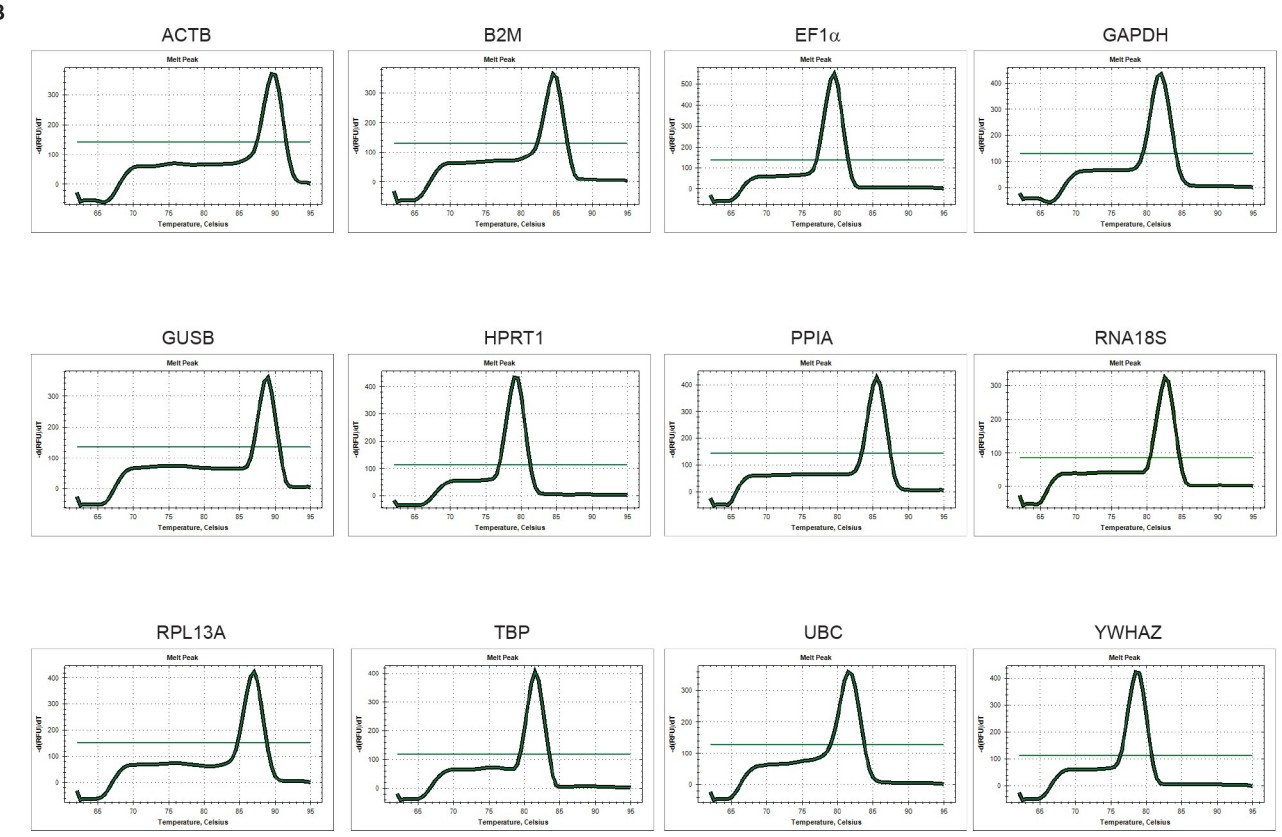

**Fig 2. Validation of primer quality.** (A) RT-qPCR products amplified with each pair of primers revealed a single product of the expected size, while the negative control did not result in any PCR product. (B) Melting curves of each pair of primers show individual dissociation at temperatures > 75˚C for all products.

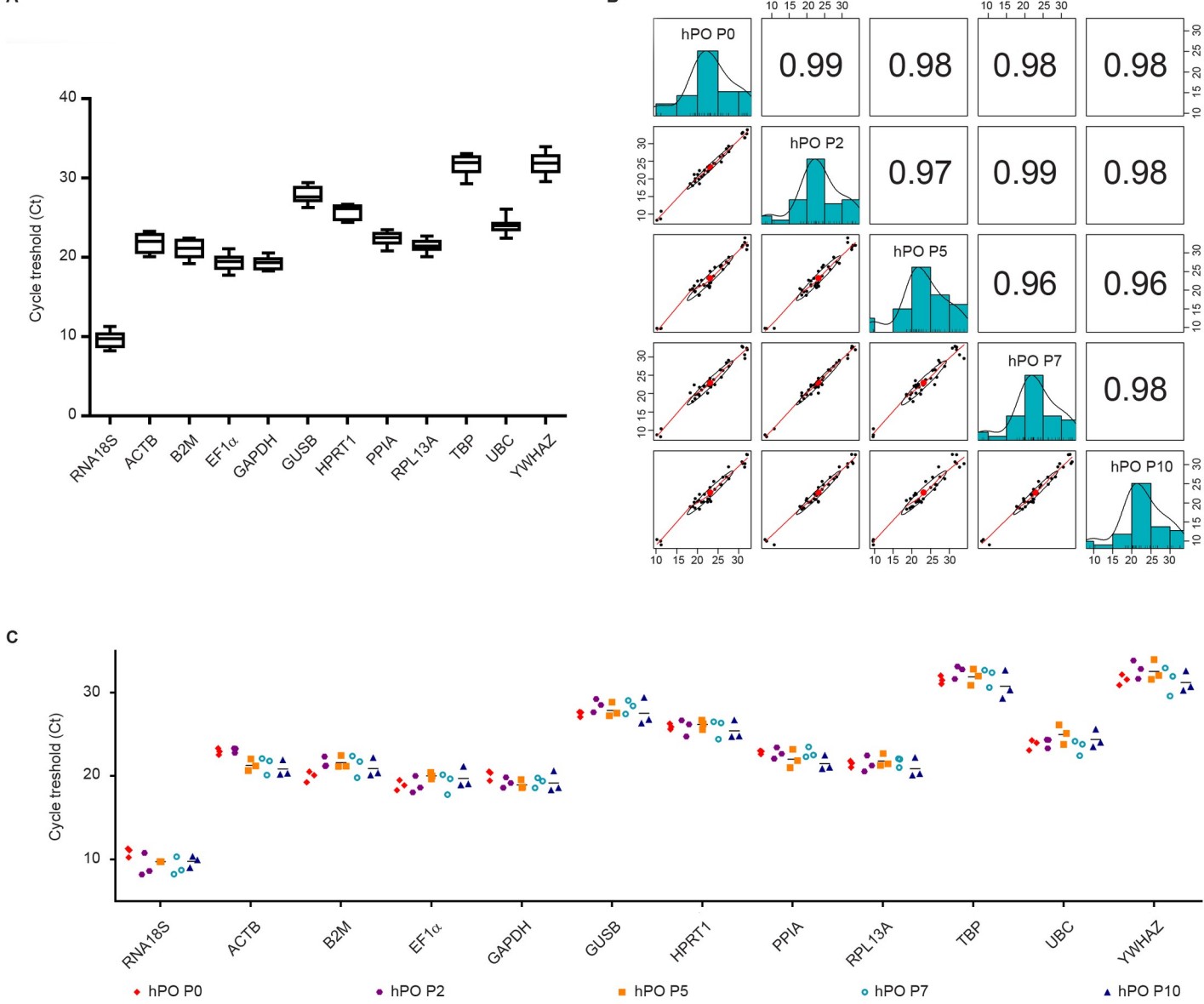

**Fig 3. Gene expression stability and pairwise correlation for reference genes.** (A) Expression data are displayed as Ct values for each reference gene in all samples. The line across the box depicts the median. The box indicates the 25th and 75th percentiles. Bars represent the maximum and minimum values. (B) Pairwise correlation analysis for each sample with all reference genes. The histograms show the median distribution of gene expression in each sample. Scatter plots show the distribution of all gene expression for each pair of samples (P0 vs P2, P0 vs P5, P0 vs P7, P0 vs P10, P2 vs P5, P2 vs P7, P2 vs P10, P5 vs P7, P5 vs P10, P7 vs P10) with the regression line and its $R^2$ value $\geq$ 0.96. (C) Expression data are displayed as Ct values for each reference gene in each sample considered. The line across the box depicts the median.

instability in the expression of the selected ductal genes was observed, and this is in contrast with the protein levels detected in our hPOs (S1 Fig). Moreover, focusing on the less stable HKG (ACTB) an apparent reduction of EpCAM and SOX9 expression was noticed, this behaviour was also in contrast with the detected protein levels (S1 Fig). Taken together, these results demonstrate that the selection and the number of appropriate HKGs to be used are critical parameters to avoid false results and to obtain reliable and significant data by gene expression analysis.

**Table 1. Ct values of candidate HKGs in pancreatic tissue, early-passage hPOs and late-passage hPOs after spike-in normalization.**

| | | YWHAZ | RPL13A | PPIA | B2M | GAPDH | TBP | ACTB | 18S | GUSB | EF1a | UBC | HPRT |
|---|---|---|---|---|---|---|---|---|---|---|---|---|---|
| **P0 vs P2** | mean | 32,14 | 21,42 | 22,78 | 20,75 | 19,65 | 31,99 | 23,01 | 10,03 | 27,93 | 18,88 | 23,87 | 25,87 |
| | st. dev. | 1,06 | 0,65 | 0,46 | 1,06 | 0,74 | 0,80 | 0,34 | 1,32 | 0,78 | 0,75 | 0,56 | 0,67 |
| | pvalue | 0,168 | 0,924 | 0,644 | 0,027 | 0,217 | 0,226 | 0,573 | 0,108 | 0,192 | 0,989 | 0,103 | 0,930 |
| **p0 vs P5** | mean | 32,01 | 21,61 | 22,42 | 20,75 | 19,52 | 31,67 | 22,10 | 10,30 | 27,64 | 19,45 | 24,37 | 26,02 |
| | st. dev. | 1,05 | 0,58 | 0,86 | 1,09 | 0,82 | 0,72 | 1,03 | 0,72 | 0,63 | 0,77 | 1,07 | 0,43 |
| | pvalue | 0,313 | 0,658 | 0,374 | 0,088 | 0,213 | 0,682 | 0,038 | 0,068 | 0,461 | 0,151 | 0,230 | 0,637 |
| **p0 vs P7** | mean | 31,49 | 21,56 | 22,80 | 20,61 | 19,67 | 31,68 | 22,12 | 9,98 | 27,85 | 19,04 | 23,60 | 25,81 |
| | st. dev. | 1,16 | 0,46 | 0,43 | 1,20 | 0,72 | 0,80 | 1,13 | 1,25 | 0,72 | 0,91 | 0,72 | 0,77 |
| | pvalue | 0,967 | 0,229 | 0,739 | 0,306 | 0,027 | 0,649 | 0,070 | 0,089 | 0,255 | 0,753 | 0,749 | 0,837 |
| **p0 vs P10** | mean | 31,34 | 21,15 | 22,16 | 20,41 | 19,63 | 31,12 | 21,86 | 10,32 | 27,46 | 19,29 | 24,06 | 25,65 |
| | st. dev. | 0,90 | 0,84 | 0,96 | 0,98 | 1,01 | 1,21 | 1,33 | 0,82 | 1,08 | 0,98 | 0,86 | 0,79 |
| | pvalue | 0,573 | 0,379 | 0,091 | 0,146 | 0,308 | 0,457 | 0,068 | 0,248 | 0,957 | 0,197 | 0,481 | 0,413 |
| **P2 vs P5** | mean | 32,63 | 21,59 | 22,35 | 21,57 | 19,07 | 32,17 | 22,19 | 9,46 | 28,14 | 19,44 | 24,48 | 26,00 |
| | st. dev. | 1,07 | 0,81 | 0,91 | 0,62 | 0,54 | 0,87 | 1,11 | 0,92 | 0,81 | 0,93 | 0,99 | 0,74 |
| | pvalue | 0,863 | 0,732 | 0,560 | 0,995 | 0,653 | 0,569 | 0,021 | 0,555 | 0,597 | 0,291 | 0,331 | 0,757 |
| **P2 vs P7** | mean | 32,11 | 21,54 | 22,73 | 21,43 | 19,22 | 32,18 | 22,21 | 9,13 | 28,35 | 19,03 | 23,72 | 25,79 |
| | st. dev. | 1,48 | 0,73 | 0,59 | 0,95 | 0,55 | 0,93 | 1,21 | 1,12 | 0,73 | 1,04 | 0,75 | 0,97 |
| | pvalue | 0,205 | 0,499 | 0,718 | 0,746 | 0,928 | 0,092 | 0,133 | 0,778 | 0,032 | 0,656 | 0,569 | 0,664 |
| **P2 vs P10** | mean | 31,97 | 21,13 | 22,08 | 21,22 | 19,18 | 31,62 | 21,96 | 9,48 | 27,96 | 19,28 | 24,18 | 25,62 |
| | st. dev. | 1,36 | 1,00 | 0,98 | 0,91 | 0,88 | 1,54 | 1,42 | 1,03 | 1,29 | 1,10 | 0,81 | 0,98 |
| | pvalue | 0,027 | 0,128 | 0,043 | 0,145 | 0,944 | 0,174 | 0,088 | 0,669 | 0,247 | 0,078 | 0,608 | 0,436 |
| **P5 vs P7** | mean | 31,98 | 21,73 | 22,37 | 21,43 | 19,09 | 31,87 | 21,29 | 9,41 | 28,06 | 19,60 | 24,21 | 25,94 |
| | st. dev. | 1,47 | 0,62 | 0,90 | 0,98 | 0,54 | 0,94 | 0,80 | 0,78 | 0,79 | 0,95 | 1,26 | 0,84 |
| | pvalue | 0,605 | 0,907 | 0,519 | 0,842 | 0,688 | 0,989 | 0,963 | 0,403 | 0,697 | 0,464 | 0,304 | 0,697 |
| **P5 vs P10** | mean | 31,84 | 21,32 | 21,72 | 21,22 | 19,05 | 31,30 | 21,04 | 9,75 | 27,66 | 19,85 | 24,67 | 25,77 |
| | st. dev. | 1,34 | 1,02 | 0,94 | 0,94 | 0,87 | 1,40 | 0,80 | 0,44 | 1,21 | 0,83 | 1,06 | 0,89 |
| | pvalue | 0,399 | 0,465 | 0,664 | 0,533 | 0,823 | 0,549 | 0,650 | 0,932 | 0,820 | 0,759 | 0,682 | 0,510 |
| **P7 vs P10** | mean | 31,32 | 21,27 | 22,10 | 21,08 | 19,20 | 31,31 | 21,06 | 9,43 | 27,88 | 19,44 | 23,91 | 25,56 |
| | st. dev. | 1,35 | 0,94 | 0,99 | 1,13 | 0,88 | 1,45 | 0,95 | 0,90 | 1,26 | 1,15 | 1,03 | 1,03 |
| | pvalue | 0,793 | 0,266 | 0,020 | 0,699 | 0,906 | 0,284 | 0,395 | 0,560 | 0,318 | 0,618 | 0,240 | 0,667 |

(P ≤ 0.05 is highlighted in yellow).

## Discussion

Three-dimensional organoid culture systems have emerged as a powerful tool to accurately recapitulate adult stem cell maintenance, differentiation and disease pathophysiology for various organs, including the brain, intestine, liver, pancreas and kidney [24–28]. Recently, our group has described and presented hPOs as a potential source of functional cells for the treatment of type 1 diabetes as they are able to be expanded and cryopreserved without morphological or molecular changes until passage 5 [9]. There is no doubt that organoids have opened remarkable opportunities very rapidly in many fields; however, to envision their use in research and in clinical contexts, from drug efficacy to organ replacement function, human organoids must be fully and extensively characterized. Due to its low cost and ease of operation, RT-qPCR is consistently used as a first step to assess the expression levels of tissue-specific genes [29, 30]. Indeed, gene expression quantification is widely considered the best method to confirm or confute the identity of any type of cell or tissue, including the emerging human organoids. The main drawbacks for this "gold standard" method are the imprecision

**Table 2. Gene expression stability rankings of the twelve reference genes calculated by NormFinder, geNorm, BestKeeper, ΔCt and RefFinder.**

| Sample | Gene | NormFinder | geNorm | BestKeeper | Δes | RefFinder | |
|---|---|---|---|---|---|---|---|
| | | Stability value | M value | ST.DEV | ST.DEV | Value | Rank |
| Human Panctreatic Organoids (Passages from 0 to 10) | ACTB | 0,684 | 0,89 | 0,95 | 1,28 | 11,24 | 12 |
| | B2M | 0,435 | 0,69 | 0,83 | 0,91 | 6,59 | 8 |
| | EF1a | 0,506 | 0,67 | 0,78 | 0,92 | 6,24 | 7 |
| | GAPDH | 0,380 | 0,69 | 0,64 | 0,93 | 4,23 | 4 |
| | GUSB | 0,334 | 0,63 | 0,80 | 0,78 | 3,46 | 3 |
| | HPRT | 0,210 | 0,65 | 0,67 | 0,73 | 1,73 | 2 |
| | PPIA | 0,416 | 0,72 | 0,68 | 0,88 | 5,69 | 5 |
| | RNA18S | 0,711 | 1,28 | 0,79 | 1,50 | 10,49 | 11 |
| | RPL13A | 0,247 | 0,61 | 0,62 | 0,73 | 1,19 | 1 |
| | TBP | 0,403 | 0,72 | 0,90 | 0,85 | 6,12 | 6 |
| | UBC | 0,515 | 0,73 | 0,67 | 0,98 | 7,21 | 9 |
| | YWHAZ | 0,493 | 0,81 | 0,94 | 0,95 | 9,43 | 10 |

and the high number of technical errors that could affect the reproducibility and the accuracy of the results generated using this technique [11].

To start the molecular characterization, some crucial aspects must be kept in mind. The first one is the appropriate choice of the reference genes to correct for errors in sample quantification as well as sample-to-sample differences in the efficiency of reverse transcription and PCR amplification [31]. The use of non-validated reference genes can lead to erroneous and inconsistent results, which are biologically meaningless. Therefore, it is important that the expression level of the selected HKG is stable and unaffected by the experimental conditions used in the analysis.

Recently, numerous studies focusing on this critical aspect [32, 33] have been presented in the literature, but not even one refers to the human organoid context. Indeed, to the best of

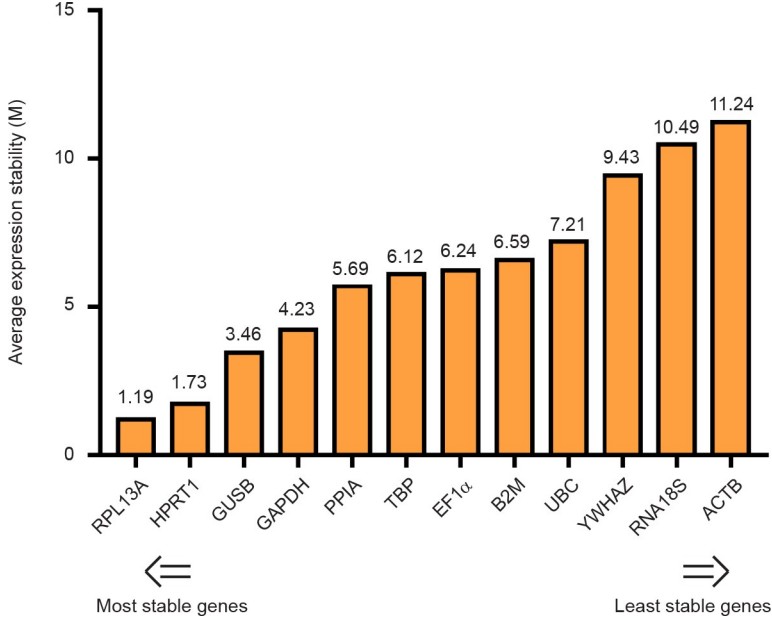

**Fig 4. Final stability ranking of the twelve reference genes as calculated by RefFinder.** The comprehensive final ranking for all samples pooled together was obtained using the RefFinder tool based on the geometric mean of the geNorm, NormFinder, BestKeeper and ΔCt method values.

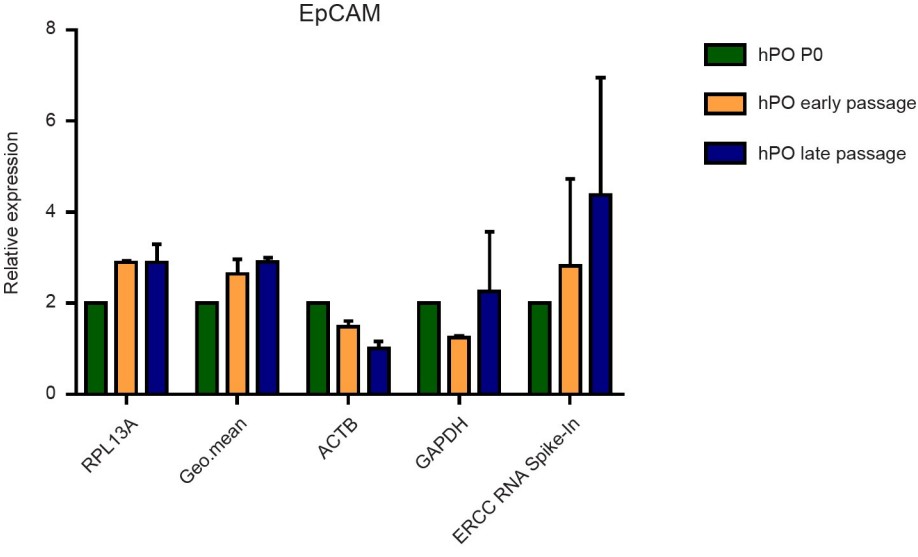

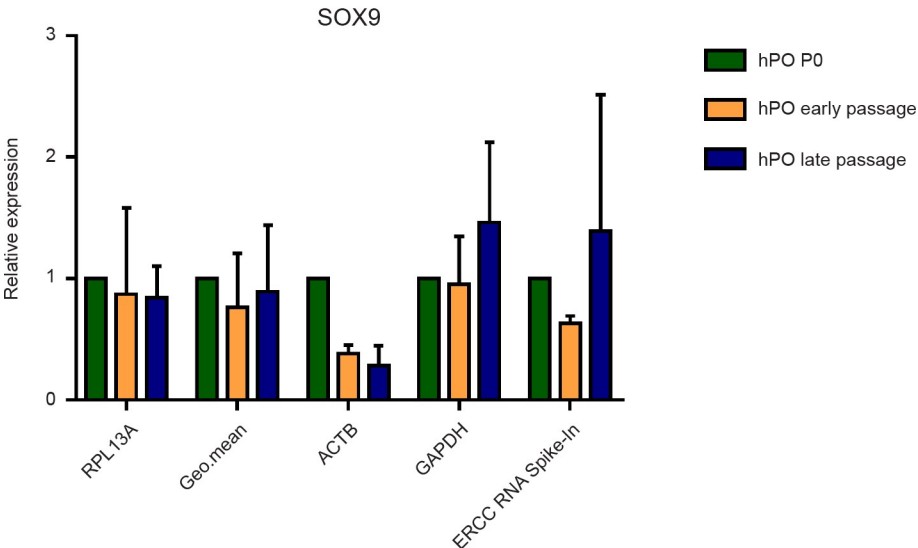

**Fig 5. Reliability of selected housekeeping genes.** The relative transcriptional level of the two ductal markers EpCAM and SOX9 in each sample condition considered was normalized with the best and the worst reference gene (RPL13A and ACTB, respectively), the geometric mean (Geo.mean) of two selected HKGs by NormFinder (PPIA and UBC) and one commonly used gene (GAPDH). One-way analysis of variance followed by Dunnett's post-hoc test for multiple comparisons was used.

our knowledge, there are no previous reports regarding the validation of reference genes for gene expression assays of hPOs. In the current work, we present the first study on hPOs focusing on the identification and validation of a suitable set of HKGs to use in gene expression analysis during short- and long-term culture. To select the best reference gene for hPO transcriptional analysis, we performed a comparison of 12 commonly used HKGs belonging to 5 different functional families: metabolism/cell viability-related genes (GAPDH, HPRT and YWHAZ); a structure/cytoskeleton-related gene (ACTB); protein folding and modification-

related genes (GUSB, PPIA and UBC); transcription/translation-related genes (EF1A, RNA18S, RPL13A and TBP); and an antigen processing-related gene (B2M). In order to identify the most suitable reference gene, we analysed the expression data using four independent statistical algorithms previously reported: BestKeeper, NormFinder, geNorm and the comparative ΔCt method. All these algorithms gave consistent results, although the most stable genes were not listed in the same order. Next, to provide a comprehensive ranking of the HKGs, we combined all of the data obtained with the four algorithms using the tool RefFinder [34]. This approach can produce precise results by integrating and balancing all of the features of the mentioned methods [13].

To evaluate the transcription levels of the selected HKGs, we first performed a preliminary validation of our primers. Our results showed no dimer formation and an appreciable efficiency (efficiency = 95–105%) combined with an acceptable level of expression (Ct < 35). Altogether, these parameters should be underlined because they play a key role for the validity of this type of study [11].

Based on our statistical analysis, we found the most stable gene to use alone or in combination for normalization of RT-qPCR analysis of hPO isolation and maintenance in culture. Interestingly, to characterize the long-term hPO culture by gene expression analysis, we observed that the most stable HKG was RPL13A, while NormFinder suggested PPIA-UBC as the best couple genes to perform data normalization. This result was in accordance with previous work showing that the mRNA expression levels of RPL13A was stable in a variety of human tissues, including pancreatic cells, making it an excellent reference genes [35–37]. On the other hand, ACTB and RNA18S, two of the most-used reference genes, appeared to be less reliable. This is not surprising, since there are many documented reports showing that the mRNA levels of ACTB and RNA18S fluctuate [38]. However, they are still commonly used as reference genes.

Furthermore, our analysis showed that the intragroup and intergroup comparisons could influence the choice of the HKG to be used. Therefore, it is evident that the identification of a stable reference gene for accurate and reproducible RT-qPCR data normalization remains a critical task, particularly when performing gene expression profile analysis using samples from different individuals.

## Conclusion

The present study is the first report of a systemic evaluation of potential HKGs suitable for accurate RT-qPCR normalization of hPOs as well as comparative studies with pancreatic tissue. The HKGs were ranked according to their stability in different experimental setups. This work provides a solid basis for future gene expression analysis in the hPO field and can be taken as an example to generally expand the knowledge of this three-dimensional culture system as well as to act as a guide for future studies.

## Supporting information

**S1 Fig. Flow cytometry analysis of human pancreatic organoids.**
(TIF)

**S1 Table. Candidate HKGs.**
(DOCX)

**S2 Table. Primer sequences, amplicon length and primer efficiencies.**
(DOCX)

**S3 Table. Coefficient of variation of selected HKGs.**
(DOCX)

**S1 File. Sanger sequencing and uncropped gel images.**
(ZIP)

## Author Contributions

**Conceptualization:** Alessandro Cherubini, Francesco Rusconi.

**Data curation:** Alessandro Cherubini, Francesco Rusconi.

**Formal analysis:** Alessandro Cherubini, Francesco Rusconi.

**Funding acquisition:** Lorenza Lazzari.

**Investigation:** Alessandro Cherubini.

**Methodology:** Alessandro Cherubini, Francesco Rusconi.

**Supervision:** Alessandro Cherubini, Lorenza Lazzari.

**Validation:** Alessandro Cherubini, Francesco Rusconi.

**Writing – original draft:** Alessandro Cherubini, Francesco Rusconi, Lorenza Lazzari.

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
