## [Decision Letter · Decision Letter 0]

6 Sep 2021

PONE-D-21-24295Human pancreatic organoids: correct choice of housekeeping gene leads to accurate and reproducible gene expression profile analysisPLOS ONE

Dear Dr. Lazzari,

Thank you for submitting your manuscript to PLOS ONE. After careful consideration, we feel that it has merit but does not fully meet PLOS ONE’s publication criteria as it currently stands. Therefore, we invite you to submit a revised version of the manuscript that addresses the points raised during the review process.

Your paper has been reviewed by two experts in the field, both of whom had found merit in your study. To improve the paper, early and late time points need to be clarified. Also note that the use of islet-depleted pancreatic tissue to compare changes of housekeeping genes in early and late organoids is somewhat misleading since the pancreatic tissue consists mostly of acinar cells, whereas organoids are almost exclusively ductal cells. 

We look forward to receiving your revised manuscript.

Kind regards,

Zoltán Rakonczay Jr., M.D., Ph.D., D.Sc.

Academic Editor

PLOS ONE

Journal Requirements:

a) Did participants provide their written or verbal informed consent to participate in this study?

 [This work was funded by the grant “LSFM4LIFE–Production and characterization of endocrine cells derived from human pancreas organoids for the cell-based therapy of Type 1 diabetes”, project number 668350.]  

Please respond by return e-mail so that we can amend your financial disclosure and competing interests on your behalf.

[hPOs were generated starting from adult healthy islet-depleted pancreatic tissue gently provided by the Diabetes Research Institute, IRCCS Ospedale San Raffaele, Milan, Italy. This work was funded by the grant “LSFM4LIFE–Production and characterization of endocrine cells derived from human pancreas organoids for the cell-based therapy of Type 1 diabetes”, project number 668350.]

 [This work was funded by the grant “LSFM4LIFE–Production and characterization of endocrine cells derived from human pancreas organoids for the cell-based therapy of Type 1 diabetes”, project number 668350.]

7. We note that you have included the phrase “data not shown” in your manuscript. Unfortunately, this does not meet our data sharing requirements. PLOS does not permit references to inaccessible data. We require that authors provide all relevant data within the paper, Supporting Information files, or in an acceptable, public repository. Please add a citation to support this phrase or upload the data that corresponds with these findings to a stable repository (such as Figshare or Dryad) and provide and URLs, DOIs, or accession numbers that may be used to access these data. Or, if the data are not a core part of the research being presented in your study, we ask that you remove the phrase that refers to these data.

Reviewers' comments:

Reviewer's Responses to Questions

**Comments to the Author**

1. Is the manuscript technically sound, and do the data support the conclusions?

Reviewer #1: Yes

Reviewer #2: Yes

2. Has the statistical analysis been performed appropriately and rigorously? 

Reviewer #1: Yes

Reviewer #2: Yes

3. Have the authors made all data underlying the findings in their manuscript fully available?

Reviewer #1: Yes

Reviewer #2: Yes

4. Is the manuscript presented in an intelligible fashion and written in standard English?

Reviewer #1: Yes

Reviewer #2: Yes

5. Review Comments to the Author

Reviewer #1: In this report, Alessandro Cherubini and colleagues present data on gene expression changes occurring during pancreas related organoid culture maturation. These results are potentially quite interesting, and it has strong scientific value which can be applied to identify the most reliable control for gene expression studies. However, they well-performed experiments to try to answer these specific questions, my feeling is that two time point limiting the evaluation of their result so i recommend additional time points to extend their study. My feeling is that although it is an interesting study with a valuable scientific background, the presented results must be supplemented with additional measurements:

The authors use an early and a late time point to measure gene expression of several housekeeping gene. I would strongly recommend including to the study the original primary cell which they used to create the organoid culture. Additionally, i also strongly suggest adding 2-3 time points between the early and late stage to prove that the development of the organoid does not affect significantly the expression of the selected genes.

Finally, i also strongly recommend changing the title form a general one to more specific-

Based on these i recommend major revision of the current study before publication.

Reviewer #2: Cherubini et al. analysed the expression of housekeeping genes in human pancreatic organoids. This is an important contribution as 3D organoid culture is an emerging technology that is utilized in a rapidly increasing number of studies. The paper is well executed and the results are analysed correctly.

Major comment:

1. The authors used the islet-depleted pancreatic tissue to compare changes of HKGs in early and late organoids. This is somewhat misleading, as the pancreatic tissue consist mostly of acinar cells, whereas organoids are almost exclusively ductal cells. Therefore comparing isolated ductal fragments with organoids seems to be more relevant.

2. How was "early' and "late" passages defined? The early passage in my point of view would be passage No. 2. In the fifth passage the organoids were grown in vitro for several weeks, which may impact gene expression. Therefore I recommend adding passage No.2. to the anaylsis as well.

MInor point:

1. The "early" and "late" organoids are described in section two in the Results, however it is already used in Figure 1.

6. PLOS authors have the option to publish the peer review history of their article (what does this mean?). If published, this will include your full peer review and any attached files.

Reviewer #1: No

Reviewer #2: **Yes: **József Maléth

---

## [Author Response · Author response to Decision Letter 0]

20 Oct 2021

Journal Requirements:

We formatted our manuscript following the PLOS ONE's style requirements. 

a) Did participants provide their written or verbal informed consent to participate in this study?

Human pancreata were obtained from the Pancreatic Islet Processing Unit of the Diabetes Research Institute, IRCCS Ospedale San Raffaele, Milan, Italy, in the context of transplantation organ, after approval of the Institutional Review Board (National Transplant Center accredited facility IT000679). Therefore, the fresh tissue was obtained from organ donors. The use of human specimens was approved by the Ethical Committee of Fondazione IRCCS Ca’ Granda Ospedale Maggiore Policlinico n° 1982, 14th January 2020.

 [This work was funded by the grant “LSFM4LIFE–Production and characterization of endocrine cells derived from human pancreas organoids for the cell-based therapy of Type 1 diabetes”, project number 668350.] 

Please respond by return e-mail so that we can amend your financial disclosure and competing interests on your behalf.

The funders had no role in study design, data collection and analysis, decision to publish, or preparation of the manuscript. Therefore, we added the appropriate sentences as suggested.

[hPOs were generated starting from adult healthy islet-depleted pancreatic tissue gently provided by the Diabetes Research Institute, IRCCS Ospedale San Raffaele, Milan, Italy. This work was funded by the grant “LSFM4LIFE–Production and characterization of endocrine cells derived from human pancreas organoids for the cell-based therapy of Type 1 diabetes”, project number 668350.]

 [This work was funded by the grant “LSFM4LIFE–Production and characterization of endocrine cells derived from human pancreas organoids for the cell-based therapy of Type 1 diabetes”, project number 668350.]

We included the acknowledgment section in cover letter as suggested.

We generated a supporting information that contains all uncropped gels underline the picture used in our manuscript and related to Figure 2.

We added the ORCID ID of all the authors into the cover letter.

7. We note that you have included the phrase “data not shown” in your manuscript. Unfortunately, this does not meet our data sharing requirements. PLOS does not permit references to inaccessible data. We require that authors provide all relevant data within the paper, Supporting Information files, or in an acceptable, public repository. Please add a citation to support this phrase or upload the data that corresponds with these findings to a stable repository (such as Figshare or Dryad) and provide and URLs, DOIs, or accession numbers that may be used to access these data. Or, if the data are not a core part of the research being presented in your study, we ask that you remove the phrase that refers to these data.

We removed the sentence “data not shown” and we generated a supporting information containing all the sequences used in our manuscript.

We added captions related to our supporting information at the end of the paper after Reference section as reported here:

Supporting information

Fig. S1 Flow cytometry analysis of human pancreatic organoids.

Table S1 Candidate HKGs.

Table S2 Primer sequences, amplicon length and primer efficiencies.

Table S3 Coefficient of variation of selected HKGs.

Supporting information file Sanger sequencing and uncropped gel images.

Reviewer #1: In this report, Alessandro Cherubini and colleagues present data on gene expression changes occurring during pancreas related organoid culture maturation. These results are potentially quite interesting, and it has strong scientific value which can be applied to identify the most reliable control for gene expression studies. However, they well-performed experiments to try to answer these specific questions, my feeling is that two time point limiting the evaluation of their result so I recommend additional time points to extend their study. My feeling is that although it is an interesting study with a valuable scientific background, the presented results must be supplemented with additional measurements:

The authors use an early and a late time point to measure gene expression of several housekeeping gene. I would strongly recommend including to the study the original primary cell which they used to create the organoid culture. Additionally, I also strongly suggest adding 2-3 time points between the early and late stage to prove that the development of the organoid does not affect significantly the expression of the selected genes.

We thank Reviewer#1 for these comments. Following his/her indication we added the results obtained using the original primary cells from which we generated our hPOs (named P0 in our revised manuscript) after removing the islet-depleted tissue. The pancreatic tissue mainly consists of acinar cells, whereas organoids are almost exclusively composed by ductal cells and this could be considered misleading. Furthermore, to reinforce our analysis we added other time points (P0, P2, P7), and now the gap between two time points is no longer than three passages, we made it to avoid the effect of the development of hPOs on the expression of our selected genes.

Finally, I also strongly recommend changing the title form a general one to more specific-one

We are grateful for this recommendation. The new title that we proposed is: “Identification of the best housekeeping gene for RT-qPCR analysis of human pancreatic organoids.”. 

Based on these I recommend major revision of the current study before publication.

Reviewer #2: Cherubini et al. analysed the expression of housekeeping genes in human pancreatic organoids. This is an important contribution as 3D organoid culture is an emerging technology that is utilized in a rapidly increasing number of studies. The paper is well executed and the results are analysed correctly.

Major comment:

1. The authors used the islet-depleted pancreatic tissue to compare changes of HKGs in early and late organoids. This is somewhat misleading, as the pancreatic tissue consist mostly of acinar cells, whereas organoids are almost exclusively ductal cells. Therefore comparing isolated ductal fragments with organoids seems to be more relevant.

We thank Reviewer#2 for these comments. Following his/her advice we decided to remove the islet-depleted pancreatic tissue from our analysis, and we replaced it with the hPO passage 0 (P0) because it is enriched of ductal fragments from which organoids are derived as suggested in our previously published paper: “Standardized GMP-compliant scalable production of human pancreas organoids”, Dossena et al., 2020 DOI: 10.1186/s13287-020-1585-2. 

2. How was "early' and "late" passages defined? The early passage in my point of view would be passage No. 2. In the fifth passage the organoids were grown in vitro for several weeks, which may impact gene expression. Therefore I recommend adding passage No.2. to the anaylsis as well.

We are grateful for these comments that improved our manuscript. Since human pancreatic organoids could be expanded for several months, we considered at “early passage” hPO at P5 (< 2 months), and at “late passage” the hPO cultured for more than P5 (> 2 months). We reported the description of early and late passages in the caption of Figure 1. 

We totally agree with Reviewer#2 that hPO at P2 are the best representation of an “early passage”, therefore we included in our study also the hPO at P2. Moreover, to exclude that the development of the organoid did affect significantly the expression of our selected genes we added other passages. In the present form, different time points (P0, P2, P5, P7, P10) are presented, therefore now the gap between the two time points is no longer than three passages.

Minor point:

1. The "early" and "late" organoids are described in section two in the Results, however it is already used in Figure 1.

We added the definition of “early” and “late” passage in the main text, Figure 1 caption (lines 159-163 revised manuscript and lines 154-158 manuscript).

---

## [Decision Letter · Decision Letter 1]

19 Nov 2021

Identification of the best housekeeping gene for RT-qPCR analysis of human pancreatic organoids.

PONE-D-21-24295R1

Dear Dr. Lazzari,

We’re pleased to inform you that your manuscript has been judged scientifically suitable for publication and will be formally accepted for publication once it meets all outstanding technical requirements.

Kind regards,

Zoltán Rakonczay Jr., M.D., Ph.D., D.Sc.

Academic Editor

PLOS ONE

Additional Editor Comments (optional):

Reviewers' comments:

Reviewer's Responses to Questions

**Comments to the Author**

1. If the authors have adequately addressed your comments raised in a previous round of review and you feel that this manuscript is now acceptable for publication, you may indicate that here to bypass the “Comments to the Author” section, enter your conflict of interest statement in the “Confidential to Editor” section, and submit your "Accept" recommendation.

Reviewer #1: All comments have been addressed

Reviewer #2: All comments have been addressed

2. Is the manuscript technically sound, and do the data support the conclusions?

Reviewer #1: Yes

Reviewer #2: Yes

3. Has the statistical analysis been performed appropriately and rigorously? 

Reviewer #1: Yes

Reviewer #2: Yes

4. Have the authors made all data underlying the findings in their manuscript fully available?

Reviewer #1: Yes

Reviewer #2: Yes

5. Is the manuscript presented in an intelligible fashion and written in standard English?

Reviewer #1: Yes

Reviewer #2: Yes

6. Review Comments to the Author

Reviewer #1: The authors answered all of my question, therefore i recommend it for accptance and publication at PLOS One.

Reviewer #2: The authors properly addressed my comments and included the requested additional analysis to the manuscript.

7. PLOS authors have the option to publish the peer review history of their article (what does this mean?). If published, this will include your full peer review and any attached files.

Reviewer #1: **Yes: **Tibor Pankotai

Reviewer #2: **Yes: **József Maléth

---

## [Editor Report · Acceptance letter]

24 Nov 2021

PONE-D-21-24295R1 

Identification of the best housekeeping gene for RT-qPCR analysis of human pancreatic organoids. 

Dear Dr. Lazzari:

I'm pleased to inform you that your manuscript has been deemed suitable for publication in PLOS ONE. Congratulations! Your manuscript is now with our production department. 

Kind regards, 

on behalf of

Dr. Zoltán Rakonczay Jr. 

Academic Editor

PLOS ONE